# Photonic-crystal exciton-polaritons in monolayer semiconductors

Long Zhang [1], Rahul Gogna[2], Will Burg[3], Emanuel Tutuc[3] & Hui Deng [1,2]

Semiconductor microcavity polaritons, formed via strong exciton-photon coupling, provide a quantum many-body system on a chip, featuring rich physics phenomena for better photonic technology. However, conventional polariton cavities are bulky, difficult to integrate, and inflexible for mode control, especially for room-temperature materials. Here we demonstrate sub-wavelength-thick, one-dimensional photonic crystals as a designable, compact, and practical platform for strong coupling with atomically thin van der Waals crystals. Polariton dispersions and mode anti-crossings are measured up to room temperature. Non-radiative decay to dark excitons is suppressed due to polariton enhancement of the radiative decay. Unusual features, including highly anisotropic dispersions and adjustable Fano resonances in reflectance, may facilitate high temperature polariton condensation in variable dimensions. Combining slab photonic crystals and van der Waals crystals in the strong coupling regime allows unprecedented engineering flexibility for exploring novel polariton phenomena and device concepts.

[1] Physics Department, University of Michigan, 450 Church Street, Ann Arbor, MI 48109-2122, USA. [2] Applied Physics Program, University of Michigan, 450 Church Street, Ann Arbor, MI 48109-1040, USA. [3] Microelectronics Research Center, Department of Electrical and Computer Engineering, The University of Texas at Austin, Austin, TX 78758, USA. Correspondence and requests for materials should be addressed to H.D. (email: dengh@umich.edu)

Control of light-matter interactions is elementary to the development of photonic devices. Existing photonic technologies are based on weakly coupled matter-light systems, where the optical structure perturbatively modifies the electronic properties of the active media. As the matter-light interaction becomes stronger and no longer perturbative, light and matter couple to form hybrid quasi-particles—polaritons. In particular, quantum-well (QW) microcavity exciton polaritons feature simultaneously strong excitonic nonlinearity, robust photon-like coherence, and a meta-stable ground state, providing a fertile ground for quantum many-body physics phenomena[1,2] that promise new photonic technology[3]. Numerous novel types of many-body quantum states with polaritons and polariton quantum technologies have been conceived, such as topological polaritons[4–6], polariton neurons[7], non-classical state generators[8–10], and quantum simulators[11–13]. Their implementation require confined and coupled polariton systems with engineered properties, which, on one hand, can be created by engineering the optical component of the strongly coupled modes, on the other hand, is difficult experimentally using conventional polariton systems.

Conventional polariton system are based on vertical Fabry Perot (FP) cavities made of thick stacks of planar, distributed Bragg reflectors (DBRs), which have no free design parameter for mode engineering and are relatively rigid and bulky against post-processing. Different cavity structures have been challenging to implement for polariton systems as conventional materials are sensitive to free surfaces and lattice mismatch with embedding crystals. The recently emerged two-dimensional (2D) semi-conductor van der Waals crystals (vdWCs)[14,15] are uniquely compatible with diverse substrate without lattice matching[16]. However, most studies of vdWC polaritons so far continue to use FP cavities[17–23], which are even more limiting for vdWCs than for conventional materials. This is because monolayer-thick vdWCs need to be sandwiched in between separately fabricated DBR stacks and positioned very close to the cavity-field max-imum. The process is complex, hard to control, and may change or degrade the optical properties of vdWCs[22,24]. Alternatively, metal mirrors and plasmonic structures have been implemented[25–28]. They are more compact and flexible, but suffer from intrinsically large absorption loss and poor dipole-overlap between the exciton and field[25,29].

Here we demonstrate sub-wavelength-thick, one-dimensional dielectric photonic crystals (PCs) as a readily designable platform for strong coupling, which is also ultra-compact, practical, and especially well suited to the atomically thin vdWCs. Pristine vdWCs can be directly laid on top of the PC without further processing. Properties of the optical modes, and in turn the polariton modes, can be modified with different designs of the PC. We confirm polariton modes up to room temperature by measuring the polariton dispersions and mode anti-crossing in both reflectance and photoluminescence (PL) spectra. Strongly suppressed non-radiative decay to dark excitons due to the polaritonic enhancement is observed. We show that these polaritons have anisotropic polariton dispersions and adjustable reflectance, suggesting greater flexibility in controlling the exci-tations in the system to reaching vdWC-polariton condensation at lower densities in variable dimensions. Extension to more elaborate PC designs and 2D PCs will facilitate research on polariton physics and devices beyond 2D condensates.

## Results

**The system.** We use two kinds of transition metal dichalcogen-ides (TMDs) as the active media: a monolayer of tungsten dis-elenide ($WSe_2$) or a monolayer of tungsten disulfide ($WS_2$). The monolayers are placed over a PC made of a silicon-nitride (SiN) grating, as illustrated in Fig. 1a. The total thickness of the grating $t$ is around 100 nm, much shorter than half a wavelength, making the structure an attractive candidate for compact, integrated polaritonics. In comparison, typical dielectric FP cavity structures are many tens of wavelengths in size. A schematic and scanning electron microscopy images of the TMD-PC polariton device are shown in Fig. 1. More details of the structure and its fabrication are described in Methods. Since the grating is anisotropic in-plane, its modes are sensitive to both the propagation and polarization directions of the field. As illustrated in Fig. 1a, we define the direction along the grating bars as the $x$-direction,

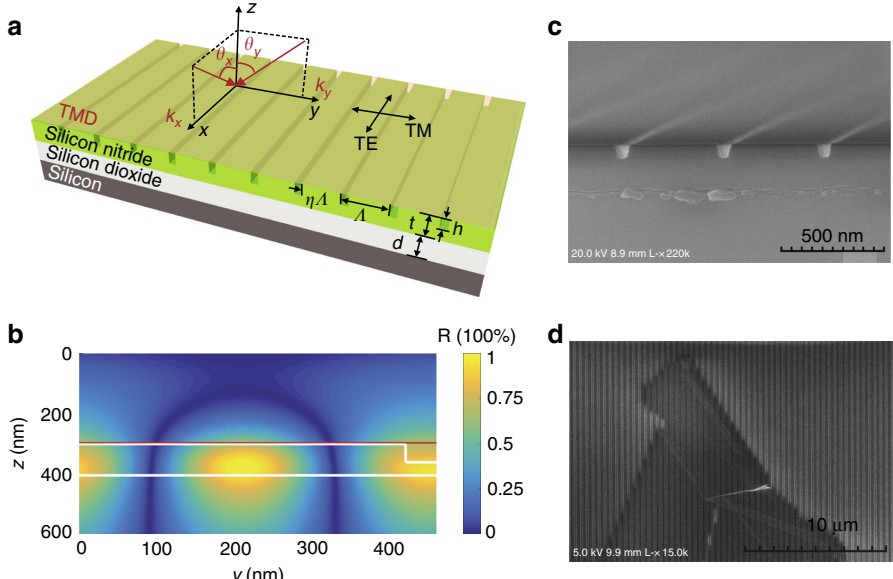

**Fig. 1** The vsWC-PC structure. **a** Schematic of the 1D PC with a monolayer TMD placed on top. The SiN PC has multiple design parameters, including the period $\Lambda$, filling factor $\eta$, total thickness $t$, and the grating thickness $h$. The $SiO_2$ capping layer has a thickness of $d$. **b** The TE-polarized electric-field profile of the PC in the $y-z$ plane. The white lines mark the outline of the PC. The red line marks the position of the monolayer TMDs. **c** A side-view SEM image of the bare PC. **d** A top-view SEM image of the TMD laid on top of a PC

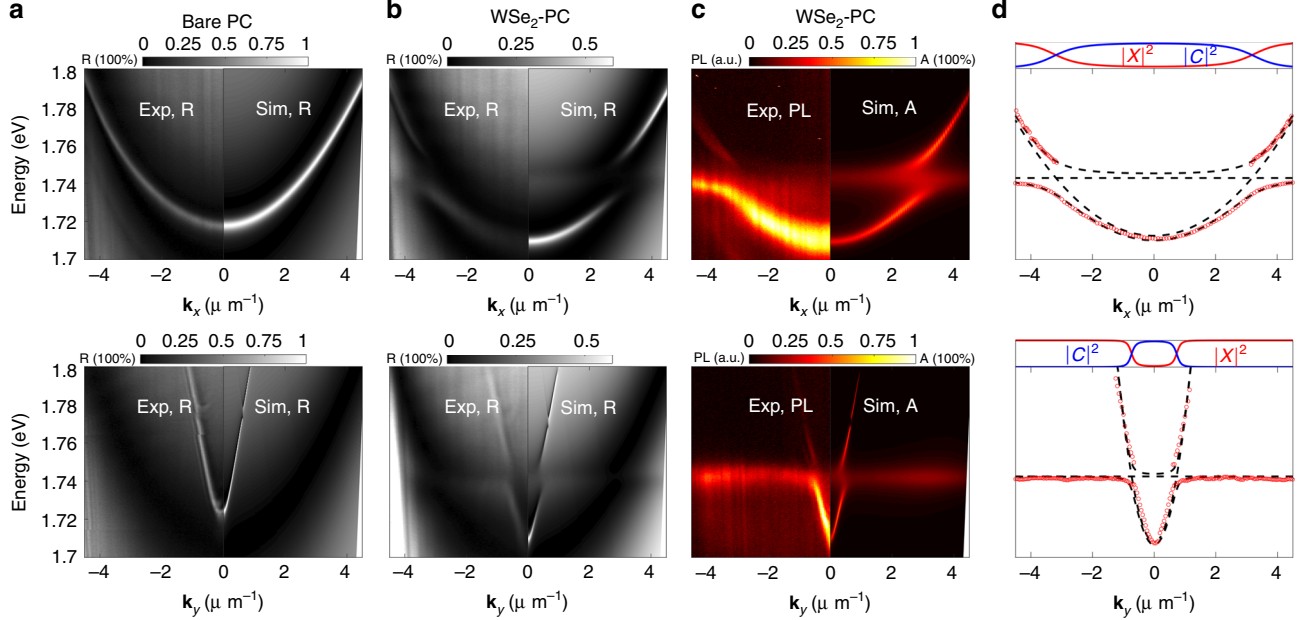

**Fig. 2** Strong coupling between TE-polarized WSe$_2$ exciton and PC modes measured by angle-resolved reflectance and PL at 10 K. The top/bottom row shows the along-bar/cross-bar directions, respectively. The left/right panels of **a**–**c** show the measured/simulated results, respectively. **a** Angle-resolved reflectance spectra of the bare PC, showing a sharp, dispersive PC mode. **b** Angle-resolved reflectance spectra of the WSe$_2$-PC integrated device, showing split, anti-crossing upper and lower polariton modes. **c** Angle-resolved PL data (left) compared with the simulated absorption spectra of the WSe$_2$-PC integrated device, showing the same anti-crossing polariton modes as in **b**. **d** The polariton energies $E_{\mathrm{LP,UP}}$ vs. wavevector $\mathbf{k}_x$, $\mathbf{k}_y$ obtained from the spectra in **c**. The lines are fits to the LP and UP dispersion with the coupled harmonic oscillator model, giving a vacuum Rabi splitting of 18.4 and 16.1 meV for the along-bar (top) and cross-bar (bottom) directions, respectively. The corresponding Hopfield coefficients $|C^2|$ and $|X^2|$, representing the photon and exciton fractions in the LP modes, respectively, are shown in the top sub-plots

across the grating bars as the $y$-direction, and perpendicular to the grating plane as the $z$-direction. For the polarization, along the grating corresponds to transverse-electric (TE), and across the bar, transverse-magnetic (TM). The TM-polarized modes are far off resonance with the exciton. Hence TM excitons remain in the weak-coupling regime, which provides a direct reference for the energies of the uncoupled exciton mode. We focus on the TE-polarized PC modes in the main text and discuss the TM measurements in the Supplementary Figure 1.

**WSe$_2$-PC polaritons.** We first characterize a monolayer WSe$_2$-PC device at 10 K. The energy-momentum mode structures are measured via angle-resolved micro-reflectance (Fig. 2a, b) and micro-PL (Fig. 2c) spectroscopy, in both the along-bar (top row) and across-bar (bottom row) directions. The data (left panels) are compared with numerical simulations (right panels), done with rigorous coupled wave analysis (RCWA).

Without the monolayer, a clear and sharp PC mode is measured with a highly anisotropic dispersion (Fig. 2a, left panels) and is well reproduced by simulation (Fig. 2a, right panels). The broad low-reflectance band in the background is an FP resonance formed by the SiO$_2$ capping layer and the substrate. The PC mode half linewidth is $\gamma_{\mathrm{cav}} = 6.5$ meV. This corresponds to a quality factor $Q$ or finesse of about 270, much higher than most TMD cavities[17,19,20,25–27] and comparable to the best DBR-DBR ones[18,21].

With a WSe$_2$ monolayer laid on top of the PC (Fig. 1c), two modes that anti-cross are clearly seen in both the reflectance and PL spectra (Fig. 2b, c) and match very well with simulations, suggesting strong coupling between WSe$_2$ exciton and PC modes. Strong anisotropy of the dispersion is evident comparing $E_{\mathrm{LP,UP}}(\mathbf{k}_x, \mathbf{k}_y = 0)$ (top row) and $E_{\mathrm{LP,UP}}(\mathbf{k}_x = 0, \mathbf{k}_y)$ (bottom row), resulting from the anisotropic dispersion of the PC modes.

Correspondingly, the effective mass and group velocity of the polaritons are also highly anisotropic, which provide new degrees of freedom to verify polariton condensation and to control its dynamics and transport properties[30].

To confirm strong coupling, we fit the measured dispersion with that of coupled modes, and we compare the coupling strength and Rabi splitting obtained from the fitting with the exciton and photon linewidth. In the strong coupling regime, the eigen-energies of the polariton modes $E_{\mathrm{LP,UP}}$ at given in-plane wavenumber $k_{\parallel}$ and the corresponding vacuum Rabi splitting $2\hbar\Omega$ are given by:

$$E_{\mathrm{LP,UP}} = \tfrac{1}{2}[E_{\mathrm{exc}} + E_{\mathrm{cav}} + i(\gamma_{\mathrm{cav}} + \gamma_{\mathrm{exc}})/2]$$

$$\pm \sqrt{g^2 + \tfrac{1}{4}[E_{\mathrm{exc}} - E_{\mathrm{cav}} + i(\gamma_{\mathrm{cav}} - \gamma_{\mathrm{exc}})]^2}, \qquad (1)$$

$$2\hbar\Omega = 2\sqrt{g^2 - (\gamma_{\mathrm{cav}} - \gamma_{\mathrm{exc}})^2/4}. \qquad (2)$$

Here $E_{\mathrm{exc}}$ is the exciton energy, $\gamma_{\mathrm{exc}}$ and $\gamma_{\mathrm{cav}}$ are the half-widths of the uncoupled exciton and PC resonances, respectively, and $g$ is the exciton-photon coupling strength. A non-vanishing Rabi splitting $2\hbar\Omega$ requires $g > |\gamma_{\mathrm{exc}} - \gamma_{\mathrm{cav}}|/2$; but this is insufficient for strong coupling. For the two resonances to be spectrally separable, the minimum mode-splitting needs to be greater than the sum of the half linewidths of the modes:

$$2\hbar\Omega > \gamma_{\mathrm{cav}} + \gamma_{\mathrm{exc}}, \text{ or, } g > \sqrt{(\gamma_{\mathrm{exc}}^2 + \gamma_{\mathrm{cav}}^2)/2}. \qquad (3)$$

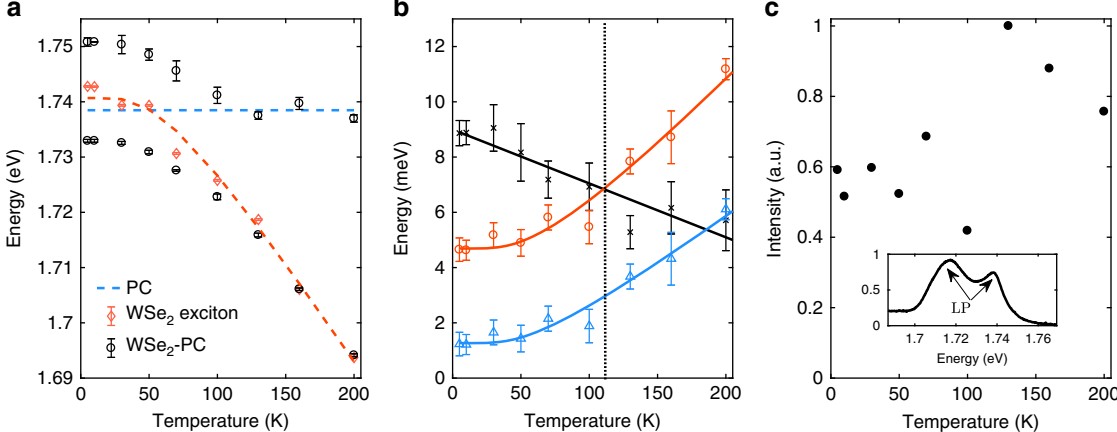

**Fig. 3** Temperature dependence of the WSe$_2$-PC system. **a** The temperature dependence of the exciton, cavity, and polariton energies. The exciton energy $E_{exc}(T)$ is measured from the weakly coupled TM excitons (red diamonds) and is extracted by fitting the PL spectrum using Gaussian line shape. Temperature dependence of the exciton is fitted by Eq. (6) (dashed blue line). The cavity energy $E_{cav}(T)$ is found to be approximately constant with temperature and indicated by the dashed blue line. The polariton energies $E_{LP,UP}(T)$ (black circles) are extracted by fitting the PL spectra at $\mathbf{k}_x = 3.1$ μm$^{-1}$, $\mathbf{k}_y = 0$ μm$^{-1}$ using Gaussian line shape. They anti-cross and split from the exciton and PC mode energies at 100 K and below, showing strong coupling in this range. All the error bars are extracted by least square data fitting. **b** The strong coupling to weak-coupling transition measured by the temperature dependence of $g$ (black stars), $\sqrt{(\gamma_{exc}^2 + \gamma_{cav}^2)/2}$ (red circles), and $(\gamma_{exc} - \gamma_{cav})/2$ (blue triangles). The red and blue lines are fits by Eq. (7) for $\gamma_{exc}$ while $\gamma_{cav}$ is approximately constant with temperature. $g$ drops to below $\sqrt{(\gamma_{exc}^2 + \gamma_{cav}^2)/2}$ at about 115 K, showing the transition to the weak-coupling regime as indicated by the dashed line. Rabi splitting persists up to about 185 K, till $g$ becomes smaller than $(\gamma_{exc} - \gamma_{cav})/2$. All the data points are extracted by fitting the PL spectrum using Gaussian line shape, and the error bars are extracted by least square data fitting. **c** Temperature dependence of the integrated PL intensity of the PC-WSe$_2$ polariton, showing only mild change in intensity. The inset shows the spectrum at 10 K integrated over $\theta_x = -30°$ to 30° and over the spectral range shown. The higher energy side shoulder corresponds to exciton-like LP emission at large $\theta_x$ where the density of states is high

In frequency domain, Eq. (3) corresponds to requiring coherent, reversible energy transfer between the exciton and photon mode. We first fit our measured PL spectra to obtain the mode dispersion $E_{LP,UP}(\mathbf{k}_{x,y})$, as shown by the symbols in Fig. 2d. We then fit $E_{LP,UP}(\mathbf{k}_{x,y})$ with (1), with $g$ and $E_{cav}(\mathbf{k}_{x,y} = 0)$ as the only fitting parameters. The exciton energy $E_{exc}$ and half-width $\gamma_{exc}$ are measured from the TM-polarized exciton PL from the same device, while the wavenumber dependence of $E_{cav}$ and $\gamma_{cav}$ are measured from the reflectance spectrum of the bare PC (Supplementary Figure 2b). We obtain $g = 8.9 \pm 0.23$ and $7.5 \pm 0.87$ meV for dispersions along $k_x$ and $k_y$, respectively, corresponding to a Rabi splitting of $2\hbar\Omega \sim 17.6$ and 14.9 meV. In comparison, $\gamma_{exc} = 5.7$ meV and $\gamma_{cav} = 3.25$ meV. Therefore $g$ is much greater than not only $(\gamma_{exc} - \gamma_{cav})/2 = 1.2$ meV but also $\sqrt{(\gamma_{exc}^2 + \gamma_{cav}^2)/2} = 4.6$ meV, which confirms the system is well into the strong coupling regime.

**Temperature dependence of WSe$_2$-PC polaritons.** At elevated temperatures, increased phonon scattering leads to faster exciton dephasing, which drives the system into the weak-coupling regime. We characterize this transition by the temperature dependence of the WSe$_2$-PC system; we also show the effect of strong coupling on exciton quantum yield.

We measure independently the temperature dependence of the uncoupled excitons via TM exciton PL, the uncoupled PC modes via reflectance from the bare PC, and the coupled modes via PL from the WSe2-PC device. We show in Fig. 3a the results obtained for $\mathbf{k}_x = 3.1$ μm$^{-1}$, $\mathbf{k}_y = 0$ μm$^{-1}$ as an example. For the uncoupled excitons, with increasing $T$, the resonance energy $E_{exc}(T)$ decreases due to bandgap reduction[31], as shown in Fig. 3a, while the linewidth $2\gamma_{exc}$ broadens due to phonon dephasing[32], as shown in Fig. 3b. Both results are very well fitted by models for conventional semiconductors (see more details in Methods). For the uncoupled PC modes, the energy $E_{cav} = 1.74$ eV and half

linewidth $\gamma_{cav} = 6.5$ meV change negligibly (Supplementary Figure 2c). The exciton and PC-photon resonances cross, as shown in Fig. 3a, at around 50 K. In contrast, the modes from the WSe$_2$-PC device anti-cross between 10 and 100 K and clearly split from the uncoupled modes, suggesting strong coupling up to 100 K. Above 130 K, it becomes difficult to distinguish the modes from WSe$_2$-PC device and the uncoupled exciton and photon modes, suggesting the transition to the weak-coupling regime.

We compare quantitatively in Fig. 3b the coupling strength $g$ with $\sqrt{(\gamma_{exc}^2 + \gamma_{cav}^2)/2}$ and $(\gamma_{exc} - \gamma_{cav})/2$ to check the criterion given in Eq. (3). The strong coupling regime persists up to about 110 K, above which, due to the increase of the exciton linewidth, $g(T)$ drops to below $\sqrt{(\gamma_{exc}^2 + \gamma_{cav}^2)/2}$ and the system transitions to the weak-coupling regime, which corresponds well to the existence/disappearance of mode-splitting in Fig. 3 below/above 110 K. On the other hand, $g > (\gamma_{exc} - \gamma_{cav})/2$ is maintained up to about 185 K. Between 110 and 185 K, coherent polariton modes are no longer supported in the structure but mode-splitting remains in the reflectance spectrum (Supplementary Figure 3).

Importantly, the temperature dependence of the polariton PL intensity reveals that strong coupling enables significant enhancement of the quantum yield of WSe$_2$ at low temperatures. It has been shown that the quantum yield of the bright excitonic states is strongly suppressed by 10- to 100-fold in bare WSe$_2$ monolayers due to relaxation to dark excitons lying at lower energies than the bright excitons[33,34]. In contrast, the WSe$_2$-PC polariton intensity decreases by less than 2-fold from 200 to 10 K. This is because coupling with the PC greatly enhances the radiative decay of the WSe$_2$ exciton-polariton states in comparison with scattering to the dark exciton states, effectively improving the quantum yield of the bright excitons.

**Room-temperature WS$_2$-PC polaritons.** To form exciton-polaritons at room temperature, we use WS$_2$ because of the

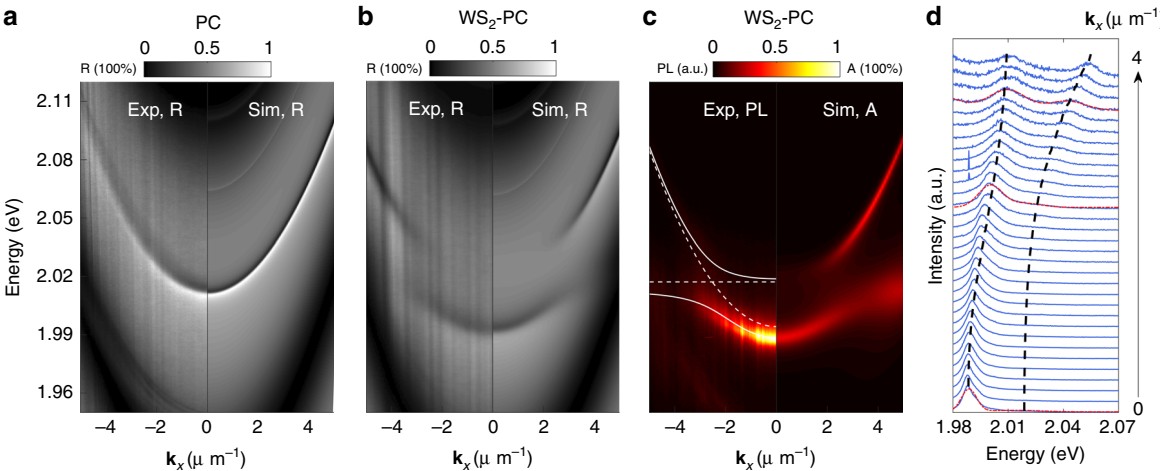

**Fig. 4** Room-temperature strong coupling between TE-polarized WS$_2$ exciton and PC modes measured by angle-resolved reflectance and PL. The left/right panels are the measured/simulated results, respectively. **a** Angle-resolved reflectance spectra of the bare PC, showing a sharp, dispersive PC mode. **b** Angle-resolved reflectance spectra of the WS$_2$-PC integrated device, showing split, anti-crossing upper and lower polariton modes. **c** Angle-resolved PL data (left) compared with the simulated absorption spectra of the WS$_2$-PC integrated device, showing the same anti-crossing polariton modes as in **b**. The solid lines are the fitted polariton dispersion, with a corresponding vacuum Rabi splitting of 22.2 meV. Dashed lines represent the exciton and cavity photon dispersion. **d** Normalized PL intensity spectra from **c** for $\mathbf{k}_x = 0\ \mu\mathrm{m}^{-1}$ (bottom) to $\mathbf{k}_x = -4\ \mu\mathrm{m}^{-1}$ (top). From Gaussian fits to the spectra (see Supplementary Figure 5 for examples), we extract the polariton energies at each wavevectors. The black dashed lines mark the fitted LP and UP positions, corresponding to the white solid lines marked in **c**

large oscillator strength to linewidth ratio at 300 K compared to WSe$_2$ (Supplementary Figure 4). We use a one-dimensional (1D) PC that matches the resonance of the WS$_2$ exciton at 300 K. The angle-resolved reflectance spectrum from the bare PC again shows a clear, sharp dispersion (Fig. 4a). The broadband background pattern is due to the FP resonance of the substrate. With a monolayer of WS$_2$ placed on top, anti-crossing LP and UP branches form, as clearly seen in both the reflectance and PL spectra (Fig. 4b, c). The data (left panels) are in excellent agreement with the simulated results (right panels). The dispersions measured from PL fit very well with the coupled oscillator model in Eq. (1), from which we obtain an exciton-photon interaction strength of $g = 12.4 \pm 0.36$ meV, above $\gamma_{\mathrm{exc}} = 11$ meV, $\gamma_{\mathrm{cav}} = 4.5$ meV, and $\sqrt{(\gamma_{\mathrm{exc}}^2 + \gamma_{\mathrm{cav}}^2)/2} = 8.4$ meV. The Rabi splitting is $2\hbar\Omega = 22.2$ meV.

**Adjustable reflectance spectra with Fano resonances**. Lastly, we look into two unconventional properties of the reflectance of the TMD-PC polariton systems: adjustable background; and Fano resonances.

As shown in Figs. 2 and 4, the reflectance around the PC polariton resonances can vary from nearly zero to close to unity. The background reflectance is determined by the FP resonances of the substrate. It features broad FP bands, with location, height, and width of the band readily adjusted by the thickness of the SiO$_2$ spacer layer, uncorrelated with the quality factor of the PC modes or the lifetime of the polaritons. For example, the WSe$_2$-PC polaritons are in the low-reflectance region of the FP bands (Fig. 5a), while the WS$_2$-PC polaritons are in the high-reflectance region (Fig. 5b). In contrast, in conventional FP cavities, high cavity quality factor dictates that the polariton modes are inside a broad high-reflectance stop-band, making it difficult to excite or probe the polariton systems at wavelengths within the stop-band. The adjustability of the reflectance in PC polariton systems will allow much more flexible access to the polariton modes and therefore may facilitate realization of polariton lasers, switches, and other polariton nonlinear devices.

Interestingly, the reflectance line shape of the PC polariton resonances are also distinctly different between the two devices (Fig. 5). A characteristic asymmetric Fano line shape is measured and varies with both the background FP band and the photon-exciton detuning. The Fano resonance arises from coupling between the sharp, discrete PC or PC polariton modes and the continuum of free-space radiation modes intrinsic to the 2D-slab structure[35]. It can be described by:

$$R = R_{\mathrm{F}}\left(\frac{(\varepsilon + q)^2}{\varepsilon^2 + 1} - 1\right) + R_{\mathrm{FP}} + I_{\mathrm{b}}, \qquad (4)$$

The first term describes the Fano resonance, where $R_{\mathrm{F}}$ is the amplitude coefficient, $q$ is the asymmetry factor, $\varepsilon = \frac{\hbar(\omega - \omega_0)}{\gamma_0}$ is the reduced energy, and $\hbar\omega_0$ and $\gamma_0$ are the resonant energy and half linewidth of the discrete mode, respectively. $R_{\mathrm{FP}}(\omega)$ and $I_{\mathrm{b}}$ are the FP background reflectance and a constant ambient background, respectively. We use the transfer matrix method to calculate $R_{\mathrm{FP}}$, then fit our data to Eq. (4) to determine the Fano parameters.

We first compare the very different Fano line shapes of the WSe$_2$-PC and the WS$_2$-PC (Fig. 5a, b). For WSe$_2$-PC, we obtain $q_{\mathrm{cav}} = 5.0$, $q_{\mathrm{LP}} = 3.5$, and $q_{\mathrm{UP}} = 4.1$, for the PC, LP, and UP modes, respectively. The large values of $q$ suggest small degrees of asymmetry and line shapes close to Lorentzian, as seen in Fig. 5a. For the WS$_2$ device, we obtain $q_{\mathrm{cav}} = 0.93$, $q_{\mathrm{LP}} = 0.83$, and $q_{\mathrm{UP}} = 0.18$, which strongly deviates from a Lorentzian line, corresponding to a more asymmetric line shape with a sharp Fano-feature, as seen in Fig. 5b. The two devices are made with gratings and substrates of different thicknesses to move the polariton resonances from the valley to the peak of the FP bands, with correspondingly a phase change of $\pi$ of the FP modes, leading to the sharp contrast between the Fano line shapes of the polaritons.

Next we focus on the WS$_2$-PC device to examine the angle-dependence of the fitted asymmetry parameter $q$ (Fig. 5c) and compare it with the polariton mode anti-crossing (Fig. 5d). As shown in Fig. 5c, the Fano asymmetry parameter $q$ for the lower

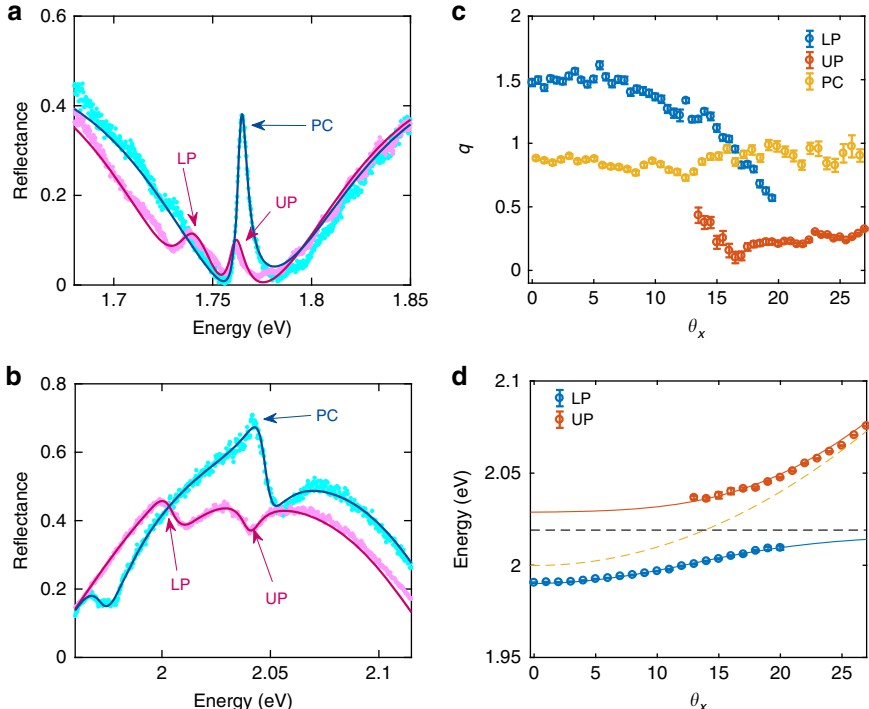

**Fig. 5** Fano resonance in the reflectance of the **a** WSe$_2$ and **b** WS$_2$ systems, measured at $\theta_x = 24°$ and 17.5°, respectively. Blue dots are the reflectance spectra of the bare PCs, and red dots, the TMD-PC devices. The lines are comparison with Fano line shape given by Eq. (4). The asymmetry parameter $q$ for PC, LP, and UP in WSe$_2$ system is 5.0, 3.5, and 4.1 respectively, and in WS$_2$ system, is 0.93, 0.83, and 0.18. **c** The fitted asymmetry parameter $q$ as a function of angle from the reflection of WS$_2$ bare PC (yellow circles) and WS$_2$-PC device (blue and red circles). **d** Extracted lower and upper polariton energies as a function of angle, as indicated by red and blue circles, respectively. Red and blue solid lines are the fittings. Yellow and black dashed lines represent the cavity and exciton dispersion. All the data points are extracted by fitting the reflection spectrum in the left panel of Fig. 4b using Eq. (4), and the error bars are extracted by least square data fitting

and upper polaritons tunes sensitively with angle near where the polaritons anti-cross (Fig. 5d), or where the polaritons are nearly half-photon and half-exciton. In contrast, for large detunings where the modes are mostly photon-like or exciton-like, as well as for the bare PC modes (yellow circles), $q$ is nearly constant over a wide range of angles. This is because, in the absence of strong exciton and photon mixing, the reflection phase of the far-detuned polaritons or the bare PC mode changes only very slowly with angle, which is also reflected in the nearly quadratic dispersion of the excitons or bare PC mode. When the PC mode strongly couples with the TMD exciton mode, the reflection phase becomes strongly dependent on the detuning and thus the angle, leading to the strong angle-dependence of the Fano line shape.

The Fano line shape sensitively depends on the phase difference between the interfering modes and therefore may enable phase-sensitive sensing applications[35,36]. In polariton systems, Fano resonances have only been reported in a ZnO microwire cavity recently[37], driven by second-harmonic generation and tunable due to phase variations of the cavity mode. Polaritons assisted in second-harmonic generation but was otherwise not necessary for forming the Fano resonance. They have not been reported in passive or linear polariton systems nor in FP cavity polaritons. Our results demonstrate Fano resonances in the polariton reflectance spectra and furthermore polariton-enabled tuning of the Fano resonances both over a wide range via the adjusting the FP resonances and finely by angle or exciton-photon detuning.

## Discussion

In short, we demonstrate integration of two of the most compact and versatile systems—atomically thin vdWCs as the active media

and PCs of deep sub-wavelength thicknesses as the optical structure—to form an ultra-compact and designable polariton system. TMD-PC polaritons were observed in monolayer WS$_2$ at room temperature and in WSe$_2$ up to 110 K, which are the highest temperatures reported for strong coupling for each type of the TMDs in dielectric cavities, respectively. The TMD-PC polaritons feature highly anisotropic energy-momentum dispersions, adjustable reflectance with sharp Fano resonances, and strong suppression of non-radiative loss to dark excitons. These features will facilitate control and optimization of polariton dynamics for nonlinear polariton phenomena and applications, such as polariton amplifiers[38], lasers[39], switches[40], and sensors[35,36].

The demonstrated quasi-2D TMD-PC polariton system is readily extended to zero-dimensional, 1D, and coupled arrays of polaritons[41,42]. The 1D PC already has many design parameter for mode engineering; it can be extended to 2D PCs for even greater flexibility. For example, 2D PCs can be designed to have chiral mode-selectivity[43,44] or to support modes of both polarizations, for controlling the spin-valley degree of freedom[45]. The TMDs can be substituted by and integrated with other types of atomically thin crystals, including black phosphorous for wide bandgap tunability[46], graphene for electrical control[47], and hexagonal boron-nitride for field enhancement.

PCs feature unmatched flexibility in optical-mode engineering, while vdWCs allow unprecedented flexibility in integration with other materials, structures, and electrical controls[48,49]. Combining the two in the strong coupling regime opens a door to novel polariton quantum many-body phenomenon and device applications[4–13].

## Methods

**Sample fabrication.** The devices shown in Fig. 1 were made from a SiN layer grown by low-pressure chemical vapor deposition on a SiO$_2$-capped Si substrate. The SiN layer was partially etched to form a 1D grating, which together with the remaining SiN slab support the desired PC modes. The grating was created via electron beam lithography followed by plasma dry etching. Monolayer TMDs are prepared by mechanical exfoliation from bulk crystals from 2D semiconductors and transferred to the grating using polydimethylsiloxane. For the WSe$_2$ device, the grating parameters are as follows: $\Lambda = 468$ nm; $\eta = 0.88$; $t = 113$ nm; $h = 60$ nm; and $d = 1475$ nm. For the WS$_2$ device, the grating parameters are as follows: $\Lambda = 413$ nm; $\eta = 0.83$; $t = 78$ nm; $h = 40$ nm; and $d = 2000$ nm.

**Optical measurements.** Reflection and PL measurements were carried out by real-space and Fourier-space imaging of the device. An objective lens with numerical aperture (NA) of 0.55 was used for both focusing and collection. For reflection, white light from a tungsten halogen lamp was focused on the sample to a beam size of 15 μm in diameter. For PL, a HeNe laser (633 nm) and a continuous-wave solid-state laser (532 nm) were used to excite the monolayer WSe$_2$ and WS$_2$, respectively, both with 1.5 mW and a 2 μm focused beam size. The collected signals were polarization-resolved by a linear polarizer then detected by a Princeton Instruments spectrometer with a cooled charge-coupled camera.

**RCWA simulation.** Simulations are carried out using an open-source implementation of RCWA developed by Pavel Kwiecien to calculate the electric-field distribution of PC modes, as well as the reflection and absorption spectra of the device as a function of momentum and energy. The indices of refraction of the SiO$_2$ and SiN are obtained from ellipsometry measurements to be $n_{SiO2} = 1.45 + \frac{0.0053}{\lambda^2}$ and $n_{SiN} = 2.0 + \frac{0.013}{\lambda^2}$, where $\lambda$ is the wavelength in the unit of μm. The WSe$_2$ and WS$_2$ monolayers were modeled with a thickness of 0.7 nm, and the in-plane permittivities were given by a Lorentz oscillator model:

$$\varepsilon(E) = \varepsilon_B + \frac{f}{E_x^2 - E^2 - i\Gamma E}. \tag{5}$$

For WSe$_2$, we used oscillator strength $f_{WS_2} = 0.7$ eV$^2$ to reproduce the Rabi splitting observed in experiments, exciton resonance $E_{WSe_2} = 1.742$ eV and full linewidth $\Gamma_{WSe_2} = 11.4$ meV based on TM exciton PL, and background permittivity $\varepsilon_{B,WSe_2} = 25$[50]. Likewise, for WS$_2$, we used $f_{WS_2} = 1.85$ eV$^2$, $E_{WS_2} = 2.013$ eV, and $\Gamma_{WS_2} = 22$ meV measured from a bare monolayer, and $\varepsilon_{B,WS_2} = 16$[51].

**Modeling the temperature dependence of the WSe$_2$ exciton energy and linewidth.** The exciton resonance energies redshift with increasing temperature as shown in Fig. 3a. It is described by the standard temperature dependence of semiconductor bandgaps[31] as follows:

$$E_g(T) = E_g(0) - S\hbar\omega\left[\coth\left(\frac{\hbar\omega}{2kT} - 1\right)\right]. \tag{6}$$

Here E$_g$(0) is the exciton resonance energy at $T = 0$ K, $S$ is a dimensionless coupling constant, and $\hbar\omega$ is the average phonon energy, which is about 15 meV in monolayer TMDs[52,53]. The fitted parameters are E$_g$(0) = 1.741 and $S = 2.2$, which agree with reported results[52,53].

The exciton linewidth $\gamma_{exc}$ as a function of temperature can be described by the following model[53,54]:

$$\gamma_{exc} = \gamma_0 + c_1 T + \frac{c_2}{e^{\hbar\omega kT} - 1}. \tag{7}$$

Here $\gamma_0$ is the linewidth at 0 K, the term linear in $T$ depicts the intravalley scattering by acoustic phonons, and the third term describes the intervalley scattering and relaxation to the dark state through optical and acoustic phonons[54]. The average phonon energy is $\hbar\omega = 15$ meV. The fitted parameters are $\gamma_0 = 11.6$ meV and $c_2 = 25.52$ meV, and $c_1$ is negligibly small in our case[53].

**Data availability.** The data that support the findings of this study are available from the corresponding author upon request.

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

## Acknowledgements

All authors acknowledge the support by the Army Research Office under Awards W911NF-17-1-0312. L.Z., R.G., and H.D. acknowledge the support by the Air Force Office of Scientific Research under Awards FA9550-15-1-0240. W.B. and E.T. acknowledge the support by National Science Foundation Grant EECS-1610008. The fabrication of the PC was performed in the Lurie Nanofabrication Facility (LNF) at Michigan, which is part of the NSF NNIN network.

## Author contributions

L.Z. fabricated the device and performed the measurements and data analysis. R.G. designed the device. W.B. and E.T. assisted in fabrication. H.D. conceived the experiment. L.Z., R.G., and H.D. wrote the paper. All authors discussed the results, data analysis, and the paper.

## Additional information

**Competing interests:** The authors declare no competing interests.

