## [Peer Review File · Nature Communications]

Reviewers' comments:

Reviewer #1 (Remarks to the Author):

A very interesting manuscript on the fabrication and characterization of a 1D photonic crystal that strongly couple with 2D transition metal dichalcogenide materials.

The effect of coupling is very clear and this opens to a new way to observe polaritons with semiconductor thin nanomaterials.

I think the authors should stress more the discussion of the Fano line shape.

They should clearly put all the parameters values of Equation 4, together to the parameters used in the transfer matrix method to calculate R_{FP} .

In the legend of Figure S2 (a) "Rflection" should be changed with "Reflection"

Reviewer #2 (Remarks to the Author):

The manuscript "Photonic-Crystal Exciton-Polaritons in Monolayer Semiconductors" by L. Zhang et al, presents experimental results on the strong coupling of light and matter in transitional metal dichalcogenide monolayers, where the confinement of light is provided by photonic crystals.

I do not feel that the manuscript merits publication in Nature Communications at present. My strongest concern is about one of the claims of the manuscript, that is, the demonstration of strong coupling at room temperature with WS₂. This is a very important claim, because operation at room temperature is important for applications.

An anticrossing is supposed to be visible in Figs. 4b and 4c (left panels, which show experimental data). However, I do not see any signature of such anticrossing in these figures. Unfortunately, I do not have the access to the original data, but I have taken the PL image 4c and attempted to perform some numerical treatment, but I still did not observe any signatures of anticrossing. I invite the authors to see for themselves the file v1.png, which is Fig4c with improved brightness/contrast. We see that the intensity maximum is clearly going upwards. There is no inflexion point, contrary to what should be observed at the anticrossing.

I am also uploading another version of the same figure, v2.png. Here I am showing the same figure with a special contour map, which allows to see the distribution of intensity better. Again, the image shows no signature of anticrossing. The branch is going upwards and decreases in intensity, which can be seen by following the contours.

If the authors indeed believe that they have an anticrossing, I would suggest them to fit the intensity emitted at each wavevector by a Gaussian and plot the position of this Gaussian, which would give the dispersion. This would prove the presence or absence of anticrossing much better than some misleading "fit" obtained by fitting something invisible. Bare photonic branch shifted down by a couple of meVs would fit the experimental intensity much better.

In fact, my overall opinion on this work is that the experimental configuration is very simple and therefore promising. An unambiguous proof of the strong coupling at room temperature would really make this paper suitable for publication in Nature Communications.

Reviewer #3 (Remarks to the Author):

The authors demonstrate polariton formation with monolayer TMD materials and a 1D photonic crystal substrate. They achieve strong coupling that extends to room temperature with WS₂. In addition to strong coupling the authors describe several features of their PC design including

anisotropic dispersion, fano resonances, and tunable reflectivity. All experimental results are thorough, well reproduced by theory, and support their understanding of the physics.

The authors' main claim seems to be that the PC design enables more freedom for modifying polariton dispersion and device realization compared to Fabry-Perot and plasmonic cavity structures. The PC design is original, but its novelty and significance is not sufficiently demonstrated by experiment. This design might find important application in the field, but the authors only allude to this direction. They do not explore any of the new physics that their design might offer.

Comments

1. As cited by the authors, other designs have achieved strong coupling both with WSe₂ and WS₂, with the later previously being achieved also at room temperature. The statement that their results show the "highest temperatures for unambiguous determination of strong coupling" is vague and should include some quantitative comparisons if this is a major part of their claim.
2. The authors devote a whole section to the temperature dependence of the strong coupling. They claim that the dark excitons are suppressed in their PC design, but only compared to bare monolayers. Shouldn't other cavity designs show similar suppression? It is unclear if the comparison to bare monolayers is meant to be supportive of their PC design or if it is simply more support for the observation of strong coupling, in which case it might not warrant its own section.
3. The anisotropic nature of the PC design could make the study of valley effects in TMD more difficult since it breaks the in-plane symmetry, which other designs do not. Considering this disadvantage the authors should be more descriptive of how they envision spin-valley control.
4. The Fano resonance that the authors observe distorts the reflectivity spectra, but does not modify the underlying polariton behavior. The authors claim that these modes could enable better addressing of polaritons within the stop band and be used to create Fano-polaritons that are tunable. However, since the authors do not perform any of these unique measurements the discussion remains speculative. Demonstrating new physics or performing a measurement that others could not would greatly enhance their claims.

Response Letter for NCOMMS-17-23725-T

"Photonic Crystal Exciton-Polaritons in Monolayer Semiconductors"

We thank the reviewers for the careful evaluation of the manuscript. We especially appreciate the constructive comments by all the reviewers that have helped improving the manuscript.

We summarize our response below followed by a detailed point-by-point response to the reviewers' comments (quoted in blue). Changes to the manuscript are underlined or **in red**.

The first reviewer recommended publication of the paper for "The effect of coupling is very clear and this opens to a new way to observe polaritons with semiconductor thin nanomaterials." The first reviewer mainly suggested to make the discussion on Fano resonance more complete. We followed the suggestion to include all parameters for the calculations of the Fano line shape and added a discussion on the first demonstration of angle tuning of the Fano line shape via strong-coupling with two new sub figures.

The second reviewer states that "my overall opinion on this work is that the experimental configuration is very simple and therefore promising. An unambiguous proof of the strong coupling at room temperature would really make this paper suitable for publication in Nature Communications." However the reviewer suggests "My strongest concern is...the demonstration of strong coupling at room temperature with WS₂". The reviewer felt anti-crossing was unclear in our WS₂ PL data and suggested more careful data analysis. We fully agree with the reviewer on performing careful analysis, and we actually performed data analysis exactly as the reviewer suggested. We have added examples and additional figures in the supplementary information to clarify any confusion. We show that anti-crossing, and therefore room temperature strong coupling, is indeed already very clearly shown in our data.

The third reviewer recognizes that "All experimental results are thorough, well reproduced by theory, and support their understanding of the physics", but thinks "The PC design is original, but its novelty and significance is not sufficiently demonstrated by experiment", and that we "do not explore any of the new physics that their design might offer." We believe we have already demonstrated the originality and significance of the PC-polariton system. We have shown it is a simple, practical, repeatable and high-quality polariton system, uniquely well suited for 2D materials. These, together with the well-established design and integration flexibility of photonic crystals, lay the ground work for the vast new opportunities our work may enable in the future. We have also shown many original and significant results for 2D-material polariton research, including: clear evidence of room-temperature strong coupling for WS₂ without the need of lossy metals, first evidence of cavity enhancement of bright excitons, highly anisotropic dispersions, adjustable reflectance background, and tunable Fano line shape. Each of them points to new research opportunities for both polaritons and 2D materials. Explorations of each of these many opportunities are where the impact of this manuscript lie in, and are beyond the scope of this first work.

Report of Reviewer #1---NCOMMS-17-23725-T

A very interesting manuscript on the fabrication and characterization of a 1D photonic crystal that strongly couple with 2D transition metal dichalcogenide materials. The effect of coupling is very clear and this opens to a new way to observe polaritons with semiconductor thin nanomaterials.

We thank the reviewer for the positive review of the quality and impact of our work.

I think the authors should stress more the discussion of the Fano line shape. They should clearly put all the parameters values of Equation 4, together to the parameters used in the transfer matrix method to calculate R_{FP} .

We thank the reviewer for the constructive suggestion. We added the complete set of parameters used to fit the Fano line shapes in Fig. 5 a-b to Table I in the Supplementary Information; we also expanded the discussion on Fano resonance to include new analysis on angle tuning of the Fano line shape, illustrated in the new sub-figures Fig. 5c-d, in Section II.E: Adjustable reflectance spectra with Fano resonances. We show how the Fano line shape is tuned by angle due to strong coupling between TMDs exciton and PC, (only) near zero exciton-photon detuning, or when there is significant exciton and photon mixing.

In the legend of Figure S2 (a) “Rflection” should be changed with “Reflection”

We apologize for the spelling mistakes and have corrected it in the revised version.

Report of Reviewer #2---NCOMMS-17-23725-T

The manuscript “Photonic-Crystal Exciton-Polaritons in Monolayer Semiconductors” by L. Zhang et al, presents experimental results on the strong coupling of light and matter in transitional metal dichalcogenide monolayers, where the confinement of light is provided by photonic crystals.

I do not feel that the manuscript merits publication in Nature Communications at present. My strongest concern is about one of the claims of the manuscript, that is, the demonstration of strong coupling at room temperature with WS₂. This is a very important claim, because operation at room temperature is important for applications.

We agree with the reviewer on the importance of demonstrating strong coupling at room temperature with WS2. Our data indeed unambiguously show strong coupling at room temperature as we explain in detail below. We hope the explanation and additional figures will fully address the reviewer’s concern.

An anticrossing is supposed to be visible in Figs. 4b and 4c (left panels, which show experimental data). However, I do not see any signature of such anticrossing in these figures.

Anticrossing is indeed visible and clear in our data, as shown in both Figs 4b and Fig. 4c, as well as in Fig. 4d.

Fig. 4b shows the raw data of k-resolved reflection spectra. As reproduced below, **both upper and lower polariton branches are clearly seen and clearly anti-cross**; it is in sharp contrast to the case without WS2, where there was clearly only one photon dispersion (as shown in Fig. 4a). The experiment is reproduced well by the simulation.

Fig. 4c shows raw data of k-resolved PL spectra, with the fitted dispersion superposed for clarity. The raw data show also the intensity distribution among different k-modes; as a result the modes with weaker intensity are less visible in the figure, which may have led to the reviewer’s skepticism. Below we normalize the PL intensity for each wave-vector to enhance the visibility at larger wavevectors (new Fig. 5Sa in the revised manuscript). Again **both anticrossing and the inflection point are very clear**. We marked the zero detuning by a red dashed line; in this case, the inflection point also fall in the region around zero-detuning, as marked by the red circle. After the inflection point, the dispersion has a negative curvature but **continues to go up in energy** toward the bare-exciton energy – which is actually as expected for polaritons.

Unfortunately, I do not have the access to the original data, but I have taken the PL image 4c and attempted to perform some numerical treatment, but I still did not observe any signatures of anticrossing. I invite the authors to see for themselves the file v1.png, which is Fig4c with improved brightness/contrast. We see that the intensity maximum is clearly going upwards. There is no inflexion point, contrary to what should be observed at the anticrossing.

I am also uploading another version of the same figure, v2.png. Here I am showing the same figure with a special contour map, which allows to see the distribution of intensity better. Again, the image shows no signature of anticrossing.

We appreciate the reviewer carefully checking the data and taking time to replot the figures. However, changing the contrast of the image file further lowers the visibility of the dispersion at larger wavevectors. We hope our figure above, with intensity normalized for each wavevector separately, clarifies any uncertainty about the inflection point. We have added this figure as Fig. S5a in the Supplementary Information.

The branch is going upwards and decreases in intensity, which can be seen by following the contours.

In fact, the lower polariton branch is **expected to be “going upwards”** in energy and **“decrease in intensity”**. It is fully consistent with and supportive of the strong-coupling regime. The decrease in intensity is expected because this is photoluminescence. Higher k modes, which are also at higher energies, should have less occupancy and lower PL intensity. The fact that we measured higher intensity near $k=0$, away from exciton-gain maximum, is actually another indication the system is indeed in the strong coupling regime, and there is thermal relaxation among the lower polaritons. If the system were in the weak-coupling regime, maximum intensity should be where the exciton mode and photon mode cross, at higher k and close to the inflection point.

If the authors indeed believe that they have an anticrossing, I would suggest them to fit the intensity emitted at each wavevector by a Gaussian and plot the position of this Gaussian, which would give the dispersion. This would prove the presence or absence of anticrossing much better than some misleading “fit” obtained by fitting something invisible. Bare photonic branch shifted down by a couple of meVs would fit the experimental intensity much better.

We are confident we have anticrossing and strong coupling. We fully agree with the reviewer on the data analysis procedure and in fact **that is exactly what we did**. The white dispersion lines plotted in Fig 4c. is NOT some hand-drawn curves fit to something invisible. They were obtained rigorously by fitting the spectra (as shown in the examples in Fig. 4d and below) to obtain the LP and UP energies at different wavenumbers, then fitting these energy vs. wavenumber by the polariton dispersion equation. **It is the same procedure the reviewer suggested, and the same procedure we followed for both WSe2 and WS2**. We already showed the fitted energies and corresponding fitted dispersions in Fig. 2d for WSe2, so we decided not to use Fig. 4d to show it again for WS2. We plotted only the final fitted dispersions for PC-WS2, in Fig. 4c, overlaid on the 2D images, and plotted in Fig. 4d the line-spectra instead to provide more data and information.

Below are some examples of the spectra and peak-fitting at different wavenumbers (new Fig. S5 c-e). We note that, in all previous room-temperature TMD polariton systems, it has not been possible to see both LP and UP photoluminescence simultaneously, let alone to perform the careful fitting and dispersion measurement as we were able to for WS2 at 300 K.

From these spectral fit, we obtain the resonance energies at each wavenumber, which is plotted in the figure to the right (symbols, new Fig. 5Sb). We then fit the two experimentally measured dispersion curves to the polariton dispersion relation. The fitted dispersions are shown as solid lines in the figure (new Fig. 5Sb) -- these two fitted upper and lower dispersions are what we plotted on the 2D image in Fig. 4c. We now add these figures – the 2D k-space PL image normalized for each wavenumber, the examples spectra with Gaussian fits, and the dispersion data obtained from the spectral fits – to Fig. S5 a-e in the Supplementary Information.

In fact, my overall opinion on this work is that the experimental configuration is very simple and therefore promising. An unambiguous proof of the strong coupling at room temperature would really make this paper suitable for publication in Nature Communications.

We thank the reviewer for reckoning our system as “very simple and therefore promising” and recommending its publication in Nature Communication given unambiguous proof of strong coupling at room temperature. We hope our explanations above have clarified any ambiguity or confusion about our data and conclusion, and our results have provided convincing evidence for strong coupling of WS2 at room temperature.

Report of Reviewer #3---NCOMMS-17-23725-T

The authors demonstrate polariton formation with monolayer TMD materials and a 1D photonic crystal substrate. They achieve strong coupling that extends to room temperature with WS2. In addition to strong coupling the authors describe several features of their PC design including anisotropic dispersion, fano resonances, and tunable reflectivity. All experimental results are thorough, well reproduced by theory, and support their understanding of the physics.

We thank the reviewer for the very positive opinion on our data and analysis.

The authors' main claim seems to be that the PC design enables more freedom for modifying polariton dispersion and device realization compared to Fabry-Perot and plasmonic cavity structures. The PC design is original, but its novelty and significance is not sufficiently demonstrated by experiment. This design might find important application in the field, but the authors only allude to this direction. They do not explore any of the new physics that their design might offer.

The reviewer is correct with our main claim but may have underestimated its significance. One of the main challenges in many-body physics and novel device applications alike is the ability to design and/or control the properties of the system. New advancement in such ability is thus often followed by rapid progress and expansion in related subjects. That is what we hope to provide with our work for both polariton research and 2D materials research – a new polariton system based on 2D materials that is capable of taking advantage of the vast design and integration possibilities offered by photonic crystals.

Implementing the new possibilities to demonstrate ground-breaking new physics or applications, while actively pursued in my group and other groups, is beyond the scope of this first, foundational work.

We consider it is already significant to demonstrate photonic-crystal polaritons in 2D materials. It is significant for polariton research, for it is the first time we have a system with nearly unlimited possibilities for mode-engineering, without sacrificing coherence or quantum efficiency (unlike

metallic structures). It also has a lateral architecture compatible with integration. Moreover, based on TMDs, such a polariton system not only can operate at room temperature and but also enjoys the flexibility offered by the 2D van der Waals crystals, such as the variety of van der Waals crystals available, the possibility to make all sorts of heterostructures, and compatibility with substrates of different materials.

The work is significant for 2D materials research, for we show that photonic crystals are well suited for coupling with those atomically thin crystals, allowing much simpler fabrication procedure, better repeatability, and unprecedented freedom for controlling the light-matter interaction. We were able to achieve the cleanest and most clear evidence of strong coupling for WSe₂ at 100K and WS₂ at 300K, compared to other work at high temperatures. And we were able to show the cavity enhancement of bright exciton emission.

The work is also significant for photonic crystal research. Photonic crystals are very mature and well recognized for its design flexibility as an optical structure. Integration with active medium or nonlinear medium, however, has been largely limited to single defects (such as quantum dots). This is mainly because the large surface to volume ratio intrinsic to photonic crystals, which leads to low quantum yield and unstable properties for conventional extended media (e.g. III-As quantum wells). But 2D materials are no longer susceptible to this limitation, able to be used and integration with photonic crystals.

We have also shown a few interesting features of this new system, such as anisotropic dispersion, adjustable background and Fano line shape in reflection, and polarization selectivity. Each of these enables new research opportunities, as we discussed briefly in the manuscript. These and other opportunities made possible by the PC-TMD system are actively pursued by ourselves and some other groups. But it is unrealistic to expect completion and inclusion of these new research projects within the current manuscript.

Comments

1. As cited by the authors, other designs have achieved strong coupling both with WSe₂ and WS₂, with the later previously being achieved also at room temperature. The statement that their results show the "highest temperatures for unambiguous determination of strong coupling" is vague and should include some quantitative comparisons if this is a major part of their claim.

We agree with the reviewer that the statement needs some more extended discussions, which we feel could become cumbersome and distractive for the current manuscript. So we have modified the statement as follows:

"TMD-PC polaritons were observed in monolayer WS₂ at room temperature and in WSe₂ up to 110 K, which are the highest temperatures reported for ~~unambiguous determination of~~ strong-coupling for each type of TMD in dielectric cavities, respectively."

Pervious work on room temperature WSe2 or WS2 did not show direct evidence of strong coupling, as we explain below. As we discussed in more detail in Section B in the manuscript, strong coupling requires:

$$g > \sqrt{(\gamma_{exc}^2 + \gamma_{cav}^2)/2} \text{ or } \Omega > \gamma_{exc} + \gamma_{cav} \quad (1)$$

Here γ_{exc} and γ_{cav} are the half-widths of the un-coupled exciton and PC resonances respectively, g is the exciton-photon coupling strength, and Ω is the Rabi splitting.

For WSe2, there was only one work, Ref 27, that claimed higher than 100 K, yet it showed only evidence of weak coupling. It reported $2\gamma_{exc}$, $2\gamma_{cav}$, and Ω as 37.5 meV, 15 meV, and 23.5 meV, respectively, and accordingly $\Omega < \gamma_{exc} + \gamma_{cav}$, clearly in the weak-coupling regime. It is so obvious, we prefer not to get into detailed discussions on this reference in the manuscript.

For WS2, two works (Ref. 20 and 25) have claimed strong coupling at room temperature, but both used metallic structures to enhance the field strength, which led to very low quality factors of a few 10s. The large intrinsic metal loss is detrimental to coherence effects that are most interesting for polaritons. Moreover, the data analysis to show strong-coupling in these works was also incorrect or insufficient – which we discuss briefly below. But we would like to avoid such discussions in the manuscript, as they may be distractive to the main points.

- Reference 25 in the manuscript used two metallic structures to show strong coupling at room temperature.

1. The first one is a Fabry-Perot cavity with a dielectric mirror and a metal mirror. The reported parameters are $2\gamma_{exc}$, $2\gamma_{cav}$, and Ω of 28 meV, 80 meV, and 90 meV (from reflection spectrum), respectively. Although they appear to satisfy the criteria for strong-coupling, we have two concerns:

- As shown in Fig. 3a of Ref 25, duplicated to the right, The fitted upper polariton branch is obviously at a higher energy than the data indicate. So the Rabi splitting was overestimated.

- As discussed by Savona et. al. [Solid State Commun. 93, 733–739 (1995)],

reflection and transmission spectra of an FP cavity may show split-peaks/dips even if the system is in the weak-coupling regime. We also illustrated this for reflection spectra of our WSe₂-PC device, as shown in Fig. S3 of the Supplementary Information of the manuscript.

When cavity or exciton linewidth is comparable to the coupling strength, reading off “Rabi-splitting” from the reflection (or even worse, transmission) spectra, as done in the reference, can at best over-estimate Rabi-splitting and at worst mistaking weak-coupling for strong-coupling. PL would be a better indication of actual Rabi-splitting or mode anti-crossing. However, the PL spectrum in Ref. 25 showed no upper polariton emission.

2. *The second structure is a plasmonic array.* It supports TE and TM modes.
 - In the TE mode: γ_{exc} , γ_{cav} , and Ω are 14 meV, 50 meV, and 60 meV, respectively, and accordingly $\Omega < \gamma_{exc} + \gamma_{cav}$; which means the system is not in the strong coupling regime.
 - In the TM mode, γ_{exc} , γ_{cav} , and Ω are 14 meV, 18 meV, and 60 meV, respectively. The provided parameters can satisfy the strong coupling criteria. However, just extracting splitting from very blurred reflection without clear anticrossing or splitting is not very convincing, as shown below.

Duplicated figure 4(a) in the reference 25

- Reference 20 in the manuscript reported parameters γ_{exc} , γ_{cav} , and Ω as 28 meV, 15-30 meV, and 20meV to 70 meV. Some of the devices in the paper may satisfy the strong coupling criteria. However, the splitting is read off transmission spectra, which may grossly overestimate the Rabi splitting [Savona et. al., Solid State Commun. 93, 733–739 (1995)]. Also, no upper polariton emission was observed in the PL measurement.

In short, although Ref. 20 and 25 might have WS₂ devices that were actually in the strong-coupling regime, it was not directly shown by the data analysis in these papers.

- 2 The authors devote a whole section to the temperature dependence of the strong coupling. They claim that the dark excitons are suppressed in their PC design, but only compared to bare monolayers. Shouldn't other cavity designs show similar suppression? It is unclear if the comparison to bare monolayers is meant to be supportive of their PC design or if it is simply more support for the observation of strong coupling, in which case it might not warrant its own section.

First of all, the section on temperature dependence has a few important results, while the suppression of dark exciton is only one of them. It also showed the following results:

1. It provides further evidence that our system is well in the strong coupling regime. The exciton mode shifts with temperature as expected. Anti-crossing is clearly seen by temperature tuning.
2. It shows the transition from strong coupling to weak coupling driven by exciton and optical-phonon scattering at elevated temperatures for WSe₂. This is useful as higher operating temperature is an important advantage of using 2D materials for polariton research or polariton devices.
3. It also shows a region where using the incorrect (but still often used) criterion for strong coupling would mis-identify weak-coupling as strong-coupling. This is still relevant given the many recent papers using incorrect criteria or applying incorrect analysis.

Now, to answer the reviewer's question on dark exciton suppression: the reviewer is correct that other cavity designs should also show similar suppression. However, it has not been reported so far, despite many works claiming strong coupling. So our result on dark exciton suppression is actually original and new. The fact we are able to observe it also (1) provides additional confirmation of the quality of our system and additional consistency check that there is cavity enhancement of the polariton decay, and (2) suggests cavity effect as a way to study and potentially control dark excitons.

3. The anisotropic nature of the PC design could make the study of valley effects in TMD more difficult since it breaks the in-plane symmetry, which other designs do not. Considering this disadvantage the authors should be more descriptive of how they envision spin-valley control.

The anisotropic nature of 1D-PC comes with both pros and cons. While it may make the study of valley effects difficult, it may allow studies of effects with enforced valley coherence, and it opens up the experimental access to un-coupled excitons simultaneously with the polaritons.

If one is mainly interested in studying valley effects, a two dimensional PC including chiral structures can be used instead. Technically, it is straightforward to extend from 1D to 2D. Scientifically, what kind of 2D PC one would use and what type of studies one'd perform are topics

we and other groups are exploring, but would be beyond the scope of the current manuscript. For example, people have proposed all dielectric photonic crystals supporting valley-dependent state (Ref 45, 46 in the revised manuscript). We follow the reviewer's suggestion to modify the discussion as follows:

Original:

"... it can be extended to 2D PCs for even greater flexibility, such as different polarization selectivity [43] for controlling the spin-valley degree of freedom [44]."

Modified:

"... it can be extended to 2D PCs for even greater flexibility. For example, 2D PCs can be designed to have chiral mode-selectivity [45, 46] or to support modes of both polarizations, for controlling the spin-valley degree of freedom [47]."

4. The Fano resonance that the authors observe distorts the reflectivity spectra, but does not modify the underlying polariton behavior. The authors claim that these modes could enable better addressing of polaritons within the stop band and be used to create Fano-polaritons that are tunable. However, since the authors do not perform any of these unique measurements the discussion remains speculative. Demonstrating new physics or performing a measurement that others could not would greatly enhance their claims

As the reviewer summarized, the PC-polaritons have reflectance spectra with special properties, which (1) "could enable better addressing of polaritons" due to adjustable off-resonance reflectance and (2) features Fano resonances "that are tunable". Both features are clearly shown and well supported by our data (Fig. 5), and thus are not speculative. They both point to new possibilities enabled by the PC-polariton devices:

- **Adjustable off-resonance reflectance.** This is clearly shown in Fig. 5a and b. The low off-resonant reflectance shown in Fig. 5a is unique to PC-polaritons, unlike polaritons in FP cavities. Some of its implications are also apparent. For example, one can optically pump the system much more efficiently with 10s of nanometers around the resonance – in comparison to conventional Fabry-Perot cavities that always have a wide stopband around the resonance. How exactly the more efficient near-resonance pumping may affect the dynamics of the system? What phenomena will result? What other ways to make use of this unusual reflectance property? These are non-trivial, interesting research topics that are now possible to be studied.
- **Tunable Fano line shape.** Fano resonances, resulting from interference between modes, are intrinsically sensitive to the phase of these modes. Therefore they are broadly studied in various system as a candidate for applications in phase sensitive studies and sensing applications. In polariton systems, Fano resonance has only been observed in ZnO nanowire cavities when driven by second Harmonic generation. Our PC-polariton system is the first to show Fano resonance in the linear regime and more importantly, angle-tuning of the Fano asymmetry parameter via strong-coupling. The comparison in Fig. 5a

and 5b showed how the reflectance line shape of the polariton resonances can be changed drastically. In the revised manuscript, we furthermore add the first demonstration of angle tuning of the asymmetry parameter of the Fano line shape as a result of strong coupling between TMDs exciton and PC (Sec. II.E and Fig. 5c-d). We may also use these features to demonstrate new physics or perform additional unique measurements, as the reviewer suggested. We are currently engaged in such research, but these are independent endeavors on their own and are beyond the scope of the current manuscript.

REVIEWERS' COMMENTS:

Reviewer #2 (Remarks to the Author):

I have read the reports of the referees and the replies of the authors. I find that the authors have successfully replied to all criticisms. I particularly appreciate the effort that they made to support their claim of strong coupling. It is indeed quite clearly demonstrated in the present version thanks to the new figures in the supplemental material.

I think that the importance of the demonstration of the efficiency of the proposed design is indeed sufficient to merit a publication in Nature Communications.

Response Letter for NCOMMS-17-23725A

"Photonic Crystal Exciton-Polaritons in Monolayer Semiconductors"

Report of Reviewer #2--- NCOMMS-17-23725A

I have read the reports of the referees and the replies of the authors. I find that the authors have successfully replied to all criticisms. I particularly appreciate the effort that they made to support their claim of strong coupling. It is indeed quite clearly demonstrated in the present version thanks to the new figures in the supplemental material.

I think that the importance of the demonstration of the efficiency of the proposed design is indeed sufficient to merit a publication in Nature Communications.

We thank the reviewer for the positive review of the quality and impact of our work.